# Methods to achieve near-millisecond energy relaxation and dephasing times for a superconducting transmon qubit

Mikko Tuokkola [1] ✉, Yoshiki Sunada [1], Heidi Kivijärvi[1], Jonatan Albanese[1], Leif Grönberg[2], Jukka-Pekka Kaikkonen[2], Visa Vesterinen [2], Joonas Govenius [2] & Mikko Möttönen [1,2] ✉

Superconducting qubits are one of the most promising physical systems for implementing quantum computers. However, executing quantum algorithms of practical computational advantage requires further improvements in the fidelities of qubit operations, which are currently limited by the energy relaxation and dephasing times of the qubits. Here, we report our measurement results of a high-coherence transmon qubit with energy relaxation and echo dephasing times surpassing those in the existing literature. We measure a qubit frequency of 2.9 GHz, an energy relaxation time $T_1$ with a median of 425 μs and a maximum of $(666 \pm 33)$μs, and an echo dephasing time $T_2^{\mathrm{echo}}$ with a median of 541 μs and a maximum of $(1057 \pm 138)$μs. We report in detail our design, fabrication process, and measurement setup to facilitate the reproduction and wide adoption of high-coherence transmon qubits in the academia and industry.

In the past two decades, the rapid development of superconducting qubits has rendered them one of the most promising candidates for realizing large-scale quantum computers[1–6]. The transmon qubit, proposed in 2007, has become the most widely used superconducting qubit due to its simplicity and performance[7,8]. It consists of a Josephson junction and a shunt capacitor which reduces the sensitivity of the qubit to charge noise but keeps the qubit sufficiently anharmonic for realizing fast, high-fidelity gate operations.

Short coherence times have historically been a disadvantage of superconducting qubits, but the previous decade has shown continuous improvements on this challenge, leading to increased fidelities of qubit operations. The longest reported energy relaxation times $T_1$ of transmons have approached but not surpassed 400-μs median value as shown in Fig. 1 and Supplementary Table 1 based on refs. 9–16. Recently, there have also been significant improvements in transmon echo dephasing times $T_2^{\mathrm{echo}}$ with the longest reported average extending to 307 μs[10,13,14,16]. Transmon qubits with coherence times above 100 μs have also been reported, for example, in refs. 17–22. With other types of superconducting qubits such as $0-\pi$ qubits and

fluxonium qubits, energy relaxation times above a millisecond have been observed, but these qubits have either had short dephasing times or low qubit frequencies[23–25].

Here, we report our latest results on high-coherence transmon qubits with improved energy relaxation and echo dephasing times. Importantly, we describe in detail our design, fabrication process, and measurement setup with the intention to promote the reproduction of our result by other academic and industrial groups.

## Results

Figure 2 shows a microscope image of a sample which is identical to the one measured in this work. It contains four transmon qubits $Q_1$–$Q_4$, each of which couples to a coplanar waveguide resonator for readout. Qubits $Q_1$ and $Q_3$ are flux-tunable by the use of a superconducting quantum interference device (SQUID), whereas $Q_2$ and $Q_4$ are fixed-frequency qubits. The four qubit–resonator pairs are identical except for the size of the Josephson junction and the length of the resonator. The readout resonators couple to a shared Purcell filter, which reduces the energy relaxation of the qubits into the readout lines[26,27]. The

[1]QCD Labs, QTF Centre of Excellence, Department of Applied Physics, Aalto University, P.O. Box 13500, Aalto, Finland. [2]VTT Technical Research Centre of Finland Ltd. & QTF Centre of Excellence, P.O. Box 1000, Espoo, Finland. ✉e-mail: mikko.tuokkola@aalto.fi; mikko.mottonen@aalto.fi

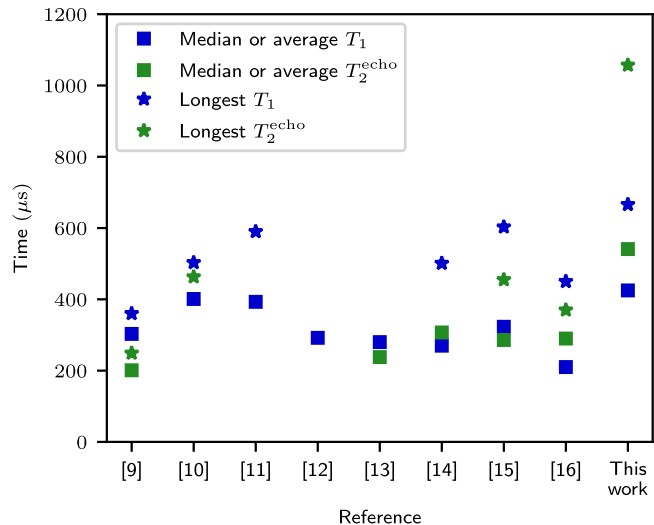

**Fig. 1 | Summary of the longest measured energy relaxation and echo dephasing times in the existing literature.** Squares represent the median or average values, and stars represent the longest values reported in the references which are chronologically ordered.

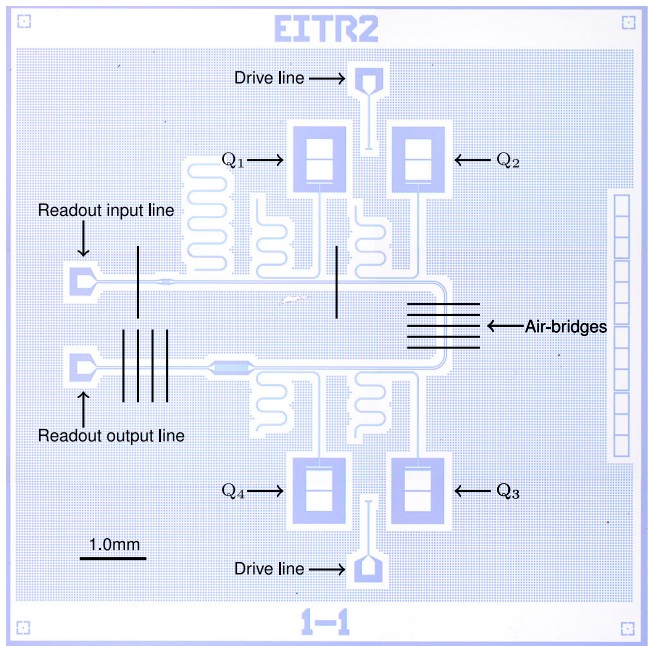

**Fig. 2 | Microscope image of a sample which is identical to the one measured in this work.** The sample contains four transmon qubits ($Q_1$–$Q_4$), their readout resonators, and a shared Purcell filter. Qubits $Q_1$ and $Q_2$ share a single drive line, and $Q_3$ and $Q_4$ share another one. Air-bridges are made by wire bonds using aluminum wires.

layout design file of the sample is available in ref. 28. For more details of the sample, see Supplementary Figs. 1 and 2.

We measure the fundamental modes of the four readout resonators to have frequencies ranging from 5.85 GHz to 6.23 GHz and the fixed-frequency qubits to have qubit frequencies of $f_{Q_2} = 2.89$ GHz and $f_{Q_4} = 3.29$ GHz. The flux-tunable qubits have maximum frequencies of $f_{Q_1} = 4.05$ GHz and $f_{Q_3} = 4.59$ GHz, which we find by sweeping the external magnetic flux applied to the sample. We then follow the standard procedure for measuring the energy relaxation time $T_1$ and echo dephasing time $T_2^{echo}$ of each qubit[2]. The flux-tunable qubits are

measured in the qubit sweet-spot, and the frequency dependence of the relaxation and dephasing times is left for future work.

During the first cooldown, we measure that the fixed-frequency qubit $Q_2$ exhibits especially long energy relaxation and echo dephasing times: medians of $T_1 = 502$ μs and $T_2^{echo}$ of 541 μs and maxima of $T_1 = (765 \pm 0.083)$ μs and $T_2^{echo} = (1057 \pm 0.138)$ μs. However, owing to the short waiting time between each measurement with the qubit $Q_2$, the qubit did not fully relax back to the ground state, resulting in unreliable $T_1$ results for the particular qubit during the first cooldown. The data for these results are shown in Fig. 3 in the form of distributions for repeated measurements of $T_1$ and $T_2^{echo}$ and time traces for the maximum values. In each $T_1$ time trace, we repeat and average 500 measurements for each value of delay time $T_{delay}$ between a π-pulse and a readout pulse. The number of repetitions is 2000 for the $T_2^{echo}$ measurement. All presented uncertainties correspond to 1σ confidence intervals obtained from the fits. The measurement results for all four qubits during the first cooldown is summarized in Table 1.

During the second cooldown, we carry out a more extensive measurement of the qubit $Q_2$. The qubit frequency is measured to have decreased by 28 MHz compared to the first cooldown. The readout resonator of this qubit is measured to have a linewidth of 2.63 MHz and a dispersive shift of $2\chi/2\pi = 1.24$ MHz. Then, we measure the energy relaxation and echo dephasing times with and without the pump signal applied to the TWPA. We fit an exponential function to the obtained data points, each of which is an average of 500 measurements, and compute the relative uncertainties of the fit parameters. We find that the TWPA does not significantly affect the coherence times but improves the certainties of the $T_1$ and $T_2^{echo}$ obtained from the fit. These results are presented in Supplementary Fig. 3. We exclude from further analysis the $T_1$ values with relative uncertainties exceeding 5% and $T_2^{echo}$ values with relative uncertainties exceeding 10%. The resulting data are shown in Fig. 3(b, c, e, f) and Supplementary Fig. 4. We obtain medians of $T_1 = 425$ μs and $T_2^{echo} = 391$ μs and maxima of $T_1 = (666 \pm 33)$ μs and $T_2^{echo} = (806 \pm 78)$ μs. The time trace for the longest measured $T_1$ deviates from an exponential function, likely because of the qubit frequency shifting during the sweep owing to the charge noise.

The data for the coherence time measurements carried out in this section are available in ref. 28. In the Supplementary Note 2 of the Supplementary Information, we demonstrate the reproducibility of the high-coherence fabrication process by measuring additional transmon qubits in three-dimensional cavity resonators.

## Discussion

In this work, we present detailed information on our design, fabrication methods, and measurement setup for a high-coherence transmon qubit. One of the four qubits on the measured sample achieved a median energy relaxation time $T_1$ of 425 μs and a median echo dephasing time $T_2^{echo}$ of 541 μs. These results surpass the previous results for a transmon qubit reported in the literature. However, the $T_1$ and $T_2^{echo}$ were significantly shorter in the second cooldown. This may be due to the redistribution of environmental fluctuators and oxidation of the sample surface. Our fabrication method and experimental setup can also be applied to other types of superconducting qubits, such as the unimon qubit, to enhance its energy relaxation and dephasing times[29], and also to large-scale manufacturing as in ref. 30.

In conclusion, this result represents a significant step in the development of high-coherence superconducting qubits by approaching the millisecond mark for the energy relaxation and dephasing times of a transmon qubit. It also demonstrates the robustness and reproducibility of the fabrication recipe described in ref. 16 and the effectiveness of the equipment and components listed in Supplementary Tables 2 and 3. Detailed reporting of a high-coherence qubit will benefit the research community and accelerate the global efforts on developing quantum sensors, quantum

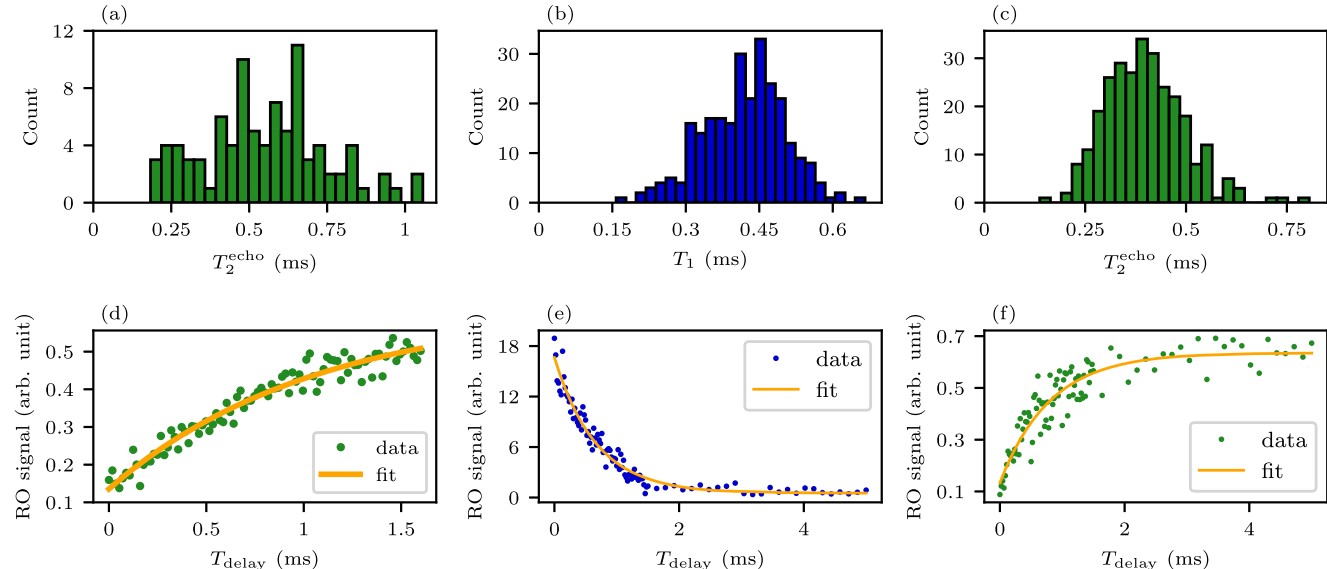

**Fig. 3 | Energy relaxation time and echo dephasing time of qubit $Q_2$. (a–c)**
Distributions of (**b**) $T_1$ and (**a**, **c**) $T_2^{\text{echo}}$ during the (**a**) first and (**b**, **c**) second cooldown. **e** Time trace for the longest measured energy relaxation time during the (**e**) second cooldown resulting in $T_1 = (666 \pm 33)\mu\text{s}$. **d, f** Time trace for the longest measured echo dephasing time during the (**d**) first and (**f**) second cooldown resulting in $T_2^{\text{echo}} = (1057 \pm 138)\mu\text{s}$ and $(806 \pm 78)\mu\text{s}$, respectively. Panels (**b**) and (**c**) include results combined with and without TWPA pump and panel (**e**) has been obtained with the pump. For other panels, the pump has not been applied.

### Table 1 | Summary of the measurement results for all four qubits during the first cooldown

| Qubit | $f_q$ (GHz) | $T_1$ (μs) | $T_2^{\text{echo}}$ (μs) | $Q$ ($\times 10^6$) |
|---|---|---|---|---|
| $Q_1$ (tunable) | 4.047 | 195 | 237 | 5.0 |
| $Q_2$ (fixed) | 2.890 | N/A[a] | 549 | 9.1 |
| $Q_3$ (tunable) | 4.593 | 44 | 32 | 1.3 |
| $Q_4$ (fixed) | 3.295 | 154 | 114 | 3.2 |

[a]During the first cooldown, $T_1$ of the qubit $Q_2$ was measured with such a short waiting time that the qubit did not decay entirely before the next measurement.

Here, $T_1$ and $T_2^{\text{echo}}$ are median values. The short $T_1$ and $T_2^{\text{echo}}$ of qubit $Q_3$ can be explained by the fact that the aluminum film evaporated on top of the resist mask inside its SQUID loop is not removed by the lift-off process.

simulators, and quantum computers based on superconducting quantum technologies.

## Methods

### Fabrication

This section provides a detailed description of our sample fabrication process, which is an adaptation of the recipe described in ref. 16 to the chemicals and equipment available to us. Notable differences include the choice of $CF_4$ as the processing gas for Nb etching, which improves the reproducibility of the etch, and the pre-dicing of the sample between the Al evaporation and lift-off, which minimizes the exposure of the sample to the ambient atmosphere. The pieces of equipment used in our process are listed in Supplementary Table 2.

**Substrate and niobium patterning.** The sample is fabricated using a 675-μs·m-thick 6-inch (100)-oriented high-resistivity ( > 10 kΩcm) intrinsic-silicon wafer sourced from Siegert Wafer. The pre-cleaning of the wafer follows ref. 31 and begins with an RCA solvent clean. The wafer then undergoes a dip in dilute hydrofluoric (HF) acid (1:100) for 1 min. After being rapidly transferred to the sputtering tool to minimize exposure to the ambient atmosphere, the wafer is baked at 300 °C under vacuum. The sputtering of a 200-nm Nb film is carried out near room temperature and at 2600-W power. The sputter target

has a purity of 99.998%. After the sputtering, the wafer is coated with a protective layer of AZ 5214E photoresist and diced into 25 mm × 30 mm rectangular coupons with a dicing saw.

The resonators, coplanar waveguides, ground plane, and transmon capacitors are patterned as described below. The selected coupon is sonicated in acetone and isopropyl alcohol (IPA) for 3 min each to remove the protective resist layer and dried with a nitrogen gun. The sample is then dehydrated on a hotplate at 110 °C for 1 min, spin-coated with AZ 5214E photoresist at 4000 rpm, and baked at 110 °C for 1 min to achieve a coating thickness of 1.4 μm. The photoresist is exposed using a maskless aligner with a laser wavelength of 405 nm and a dose of 130 mJ/cm². Subsequently, the photoresist is developed in AZ 726MIF for 1.5 min and rinsed in deionized water (DIW) for 1.5 min.

The Nb film is patterned in a plasma processing system by a chemical dry-etching process, clearing the areas revealed by the resist development step. Immediately before the process, the empty plasma chamber is cleaned with a combination of $CF_4$ and $O_2$ gases for a total of 5 min. We then load the sample into the chamber and apply oxygen plasma ashing for 10 s at 100-mTorr chamber pressure, 40-sccm gas flow, and 150-W rf source power to remove the post-development residue of photoresist on the sample surface. Then, we pump out the oxygen and introduce $CF_4$ at 50-mTorr chamber pressure, 20-sccm flow, and 30-W source power. To ensure that the Nb film is fully etched, we carry out an additional etch after unloading, visually inspecting, and re-loading the sample. The etch rate of Si is significantly lower than that of Nb, which helps us to achieve convenient control of the etch depth. The typical total etching time of Nb film is around 10 min. After the etching process, we remove some of the residual chemicals by applying another oxygen plasma ashing for 2 min without breaking the vacuum.

The photoresist is removed by immersing the sample in an $N$-methylpyrrolidone (NMP)-based solvent Remover PG at 80 °C overnight and sonicating it in the same Remover PG, acetone, and IPA for 3 min each. Subsequently, the sample is dried with a nitrogen gun and plasma-ashed again for 2 min. Using a profilometer, we measure the etch depth to be 250 nm, which implies that the Si substrate is etched by 50 nm.

**Electron-beam lithography.** In order to remove the oxide layers on the Nb and Si surfaces, we immerse the sample in dilute HF acid (0.5%) for 10 min and rinse it in DIW for 5 min[32]. The sample is then carried to a spin coater while immersed in fresh DIW to minimize its exposure to the ambient atmosphere. After spin-drying the sample, we immediately spin coat the methyl methacrylate (MMA) EL11 copolymer resist (11% solid content in ethyl lactate) at 4000 rpm, bake it at 180 °C for 5 min, and cool it for 3 min. Then we coat the sample with polymethyl methacrylate (PMMA) 950 A4 resist (950,000 molecular weight, 4% solid content in anisole) at 1000 rpm and bake it at 180 °C for 5 min. This creates a two-layer resist stack with approximately 500 nm of MMA and 400 nm of PMMA.

The resist mask for Manhattan-style Josephson junctions is patterned onto the resist stack using an electron-beam writer with 100-kV acceleration voltage, 300-µm aperture, and 0.5-nA beam current. We use a dose of 1000 µC/cm² to define the junction structure in both the PMMA layer and the MMA layer. In addition, we use a dose of 400 µC/cm² to define an undercut at each end of the line-like structures for the junctions. The undercut serves to separate the aluminum junction evaporated onto the Si substrate from the MMA side walls. We use a mask pattern very similar to that of ref. 16 to define the Josephson junctions. The electron-beam resist is developed by immersing the sample in methyl isobutyl ketone (MIBK):IPA (1:3) solvent for 5 min, rinsing it in IPA for 1 min, and drying it with a nitrogen gun.

**Junction deposition.** The Josephson junctions are deposited using an ultra-high-vacuum electron-beam evaporator with separate load-lock, oxidation, and evaporation chambers. After loading the sample, we pump the system for 14 h to reach a load-lock chamber pressure below $10^{-7}$ mbar. Subsequently, we carry out ozone ashing at 10 mbar for 1 min to remove a thin layer of resist residues.

After the ozone cleaning, the load-lock chamber is pumped again to a high vacuum ($< 10^{-7}$ mbar), and the sample is transferred to the oxidation/evaporation chamber. To fabricate the junctions, we evaporate Al at the rate of 0.2 nm/s at tilt and in-plane rotation angles specific for each of the four Al line strips as discussed below. Prior to each Al evaporation step, the oxidation and evaporation chambers are getter-pumped by evaporating Ti with a closed shutter at the rate of 0.1 nm/s for 2 min and waiting for the chamber pressure to decrease below $5 \times 10^{-8}$ mbar.

For the bottom layer of the junctions, we deposit 40 nm of Al at $\theta = 45°$ tilt and $\phi = -45°$ planetary angle. Then we create the insulating aluminum oxide layer by static oxidation at 1.2 mbar for 5 min. During the oxidation, the aluminum source is protected from the oxidation by a valve that blocks the oxygen flow. After the oxidation, we deposit the second Al layer in two steps with $\theta = 45°$ tilt and two planetary angles: first 30 nm at $\phi = 45°$ and then 30 nm at $\phi = -135°$. This ensures that both sides of the oxidized bottom strip are covered by the second Al layer.

In order to ensure a galvanic contact between the Josephson junctions and the Nb capacitor pads, we transfer the sample to the load-lock chamber and carry out argon milling with $\theta = 45°$ tilt at two planetary angles $\phi = \pm 90°$ for 2 min each at 10-sccm gas flow, 400-V beam voltage, 60-mA beam current, and 80-V acceleration voltage. This removes any Nb surface oxide that has grown in the areas exposed after the electron-beam resist development[33,34]. After the argon milling, the sample is transferred back to the oxidation/evaporation chamber, where we deposit the connecting leads between the junction and the Nb pads in three steps with 30/60/60-nm thicknesses, $\theta = 45°$ tilt and $\phi = 180°/0°/180°$ planetary angles.

As the final step, the sample is transferred back to the load-lock chamber and oxidized at 20 mbar for 10 min to create a clean oxide layer on top of the junctions before exposing the sample to the ambient atmosphere.

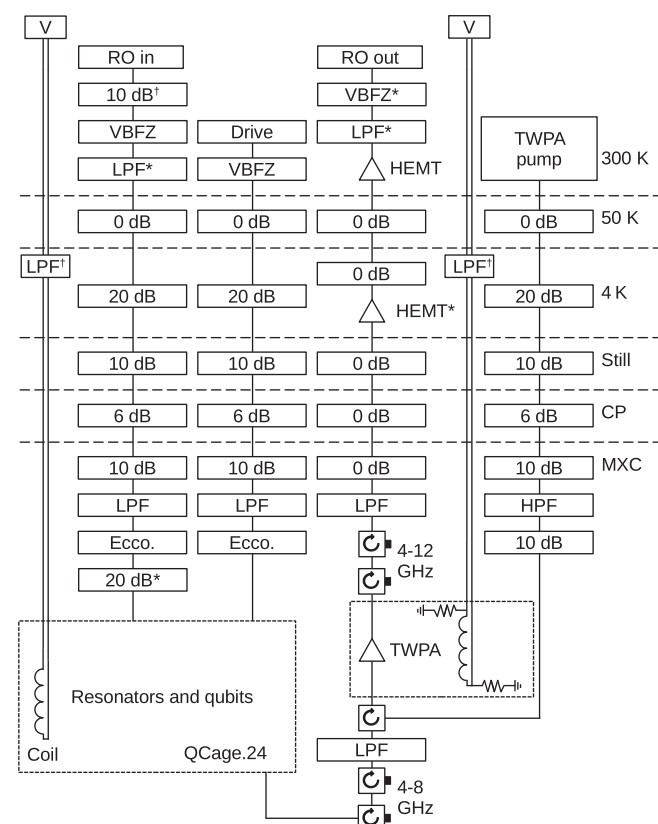

**Fig. 4 | Schematic of the experimental setup for qubit characterization and measurement.** See Supplementary Table 3 for a description of each component. For the second cooldown, the 10-dB attenuator in the drive line at the mixing-chamber (MXC) plate is removed, and the first 10-dB attenuator in the TWPA pump line at the MXC plate is replaced with an Eccosorb filter.

**Dicing and liftoff.** After the Josephson junction deposition, we coat the sample with a protective layer of AZ 5214E photoresist and pre-dice the sample with a dicing saw, cutting two-thirds deep of the substrate thickness. Before attaching the sample to the dicing tape, an ionizer fan is applied to the tape to reduce the possibility of electrostatic-discharge damage on the sample. By pre-dicing the sample prior to the liftoff, we can skip one of the resist removal steps carried out in ref. 16.

The pre-diced sample is immersed in Remover PG at 80 °C for 3 h, after which large Al flakes can be removed from the beaker using a pipette. The sample is sonicated in the same Remover PG, acetone, and IPA for 3 min each. From IPA, the sample is quickly dried with a nitrogen gun and immediately transferred to a vapor prime oven that applies a monolayer of hexamethyldisilazane (HMDS) on the sample surface, which may help to slow down the post-fabrication oxidation of the Nb and Si surfaces. The oven also has the effect of annealing the sample at 150 °C for a total of 20 min under vacuum and nitrogen environments.

The critical currents of the fabricated junctions are estimated by measuring their room-temperature resistances using a probe station. The critical currents for the sample measured in this work are significantly smaller than we had targeted, which led to low $E_J/E_C$ ratios, as low as 20 for qubit $Q_2$. The sample is then manually cleaved into separate chips utilizing the cuts established with the dicing saw, and selected chips are taken for further characterization.

### Setup for qubit measurements

Our experimental setup used for qubit characterization and measurements is presented in Fig. 4, with detailed information on measurement equipment and components provided in Supplementary

Table 3. The sample is wire-bonded to the printed circuit board (PCB) of a QCage.24 sample holder and placed inside QCage Magnetic Shielding, both of which are supplied by QDevil (under Quantum Machines). The chip is suspended by four corners inside a cavity and clamped down by the PCB. The assembly is placed inside a light-tight superconducting aluminium enclosure to reject stray microwave and infrared photons before being mounted inside the magnetic shield. The sample is exposed to ambient atmosphere at room temperature for a total of 7 days. It is then cooled down to approximately 10 mK in a Bluefors dilution refrigerator equipped with a tin-plated copper shield at 10 mK and an Amumetal magnetic shield just inside the outer vacuum chamber (OVC). We directly generate the qubit control and readout signals without any analog mixers by using a Xilinx RFSoC evaluation board with QICK firmware[35,36] locked to a Rb frequency standard. The signals then pass through rf attenuators, an Eccosorb infrared filter, and a low-pass filter before reaching the sample. The readout signal coming out of the sample goes through a three-wave-mixing Josephson traveling-wave parametric amplifier (TWPA), a cryogenic high-electron-mobility-transistor (HEMT) amplifier, and a room-temperature HEMT amplifier before being digitized by RFSoC. During the first cooldown, the TWPA is not pumped and therefore does not provide any amplification.

After the first cooldown, we reconfigure the attenuators as described in the caption of Fig. 4 before starting another cooldown. During the second cooldown, we compare measurements with and without a pump signal applied to the TWPA. The 10.5-GHz pump signal passes through a 4–12 GHz circulator, reflects from a 8-GHz low-pass filter, and travels through the circulator again before reaching the TWPA. This configuration allows us to combine the pump and readout signals without the power dissipation of a directional coupler or the impedance mismatch of a diplexer.

## Data availability
The data that support the findings of this study are available at https://doi.org/10.5281/zenodo.12819934.

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

## Acknowledgements

We acknowledge the Research Council of Finland under its Centre of Excellence Quantum Technology Finland (Grant nos. 352925 awarded to Prof. Jukka Pekola with project funding to M.M. and 352934 awarded to V.V.) and the Finnish Quantum Flagship (project funding awarded to M.M.), the European Research Council under the Advanced Grant ConceptQ (no. 101053801 awarded to M.M.), the Finnish Quantum Institute InstituteQ (PhD project funding of M.T. awarded to M.M.), Business Finland through Quantum Technologies Industrial project (no. 41419/31/ 2020 awarded to Assoc. Prof. Sorin Paraoanu with project funding to M.M.), Horizon Europe programme HORIZON-CL4-2022-QUANTUM-01-SGA through OpenSuperQPlus100 project (no. 101113946 awarded to Assoc. Prof. Sorin Paraoanu with project funding to M.M.), Jane and Aatos Erkko Foundation (project SystemQ awarded to M.M.), QDOC Doctoral Pilot Programme (PhD project funding of H.K. awarded to M.M.), and the provision of facilities and technical support by Aalto University at the OtaNano-Micronova Nanofabrication Centre.

## Author contributions

Y.S. designed the samples and cavities. Y.S. fabricated samples with help of L.G., J.-P.K., V.V. and J.G. Y.S. built the experimental setup with help from M.T. Y.S. and M.T. wrote the measurement code with help from J.A. M.T. and Y.S. conducted the experiments. M.T. analysed the data. M.T., Y.S., H.K., and M.M wrote the manuscript with comments from all authors. M.M. supervised the work with the help of J.G.

## Competing interests

M.M. declares that he is a Co-Founder and Shareholder of the quantum-computer company IQM Finland and of the quantum-algorithm company QMill Oy. J.G. and V.V. declare that they are Co-Founders and Shareholders of the quantum-hardware company Arctic Instruments Oy. Other authors declare no competing interests.
