## [Transparent Peer Review file · Nature Communications]

Methods to achieve near-millisecond energy relaxation and dephasing times for a superconducting transmon qubit

Corresponding Author: Professor Mikko Möttönen

Version 0:

Reviewer comments:

Reviewer #1

(Remarks to the Author)

I would revise the comments as follows:

I sincerely appreciate the authors' transparency and thorough disclosure of technical details. The community undoubtedly benefits from such open sharing of research. While this spirit of openness should be encouraged, it does not inherently justify publication in a particular journal. Many significant works are described in detail, but that does not automatically qualify them for publication in high-impact journals. The authors have not sufficiently demonstrated substantial novelty in their work. Journals like Nature Communications have well-established standards and criteria that must be adhered to by every submission to maintain fairness and integrity.

For instance, related works such as [9] introduced a new material platform that significantly enhanced the T1 time to 0.3ms in a planar design for the first time, while [10] further improved it to 0.5ms. This current work advances the field with slightly improved T1 time and about a factor of 2 improvement in T2 echo time but still in isolated qubits. This improvement is not substantial, and the underlying reasons are neither explained nor investigated. Even if these improvements were considered significant, they do not necessarily translate to success in multi-qubit scenarios. Not to mention that T2 echo time is not the single metric that matters. I anticipate that the authors will encounter challenges in achieving the same metrics when qubits are interconnected within a realistic and scalable architecture.

Regrettably, I must recommend rejection once more. Although I still believe this work deserves publication, it may be better suited to a more specialized journal. Before resubmission, there are several critical points that the authors need to address:

As highlighted by Reviewer #2, the transmon qubits employed in this study have a relatively low E_J/E_C ratio, which implies reduced sensitivity to dielectric loss. This could be an alternative explanation for the long T1 times, potentially overshadowing the contributions of material and fabrication factors.

It is well understood that low E_J/E_C ratios result in strong charge dispersion, which likely leads to shorter Ramsey times. The authors should not omit the Ramsey results, as they are crucial for a comprehensive understanding of the work.

This study includes measurements of tunable qubits. It is unclear why the authors did not measure qubit lifetime at varying frequencies. Such characterizations are standard in the field and could be completed in a day or two, even without rapid flux biasing, contrary to the authors' claim of it taking a year.

I strongly recommend including data on the thermal excitation of qubits and resonators. These metrics are essential for a thorough understanding of coherence times.

By addressing these points, the authors can significantly strengthen their manuscript and enhance its potential for successful publication in a suitable journal.

(Remarks on code availability)

Reviewer #2

(Remarks to the Author)

The authors have carefully addressed the comments. The additional texts and supplementary note illustrate the scope of the manuscript and the implications of the presented results. I have no further comments, except for the following specific point – the expression in line 174 may warrant attention from the authors: "a high vacuum ($<10^{-7}$ mbar) is pumped again to the load-lock chamber".

(Remarks on code availability)

Reviewer #3

(Remarks to the Author)

I have broadly reviewed the manuscript submitted by Tuokkola et al. The authors present a detailed methodology for fabricating high-quality transmon qubits.

From a technical standpoint, I find that the peer-review process has improved the quality of the manuscript to an acceptable level. However, the primary concern remains on the novelty of the work, as it is essentially a methodology-focused study.

A previous study (Ref. [16] by Kippenberg et al., Nat. Comm. 525,15,3950(2024)) reported a similar fabrication technique. In the present study, the authors employ a different combination of gases in their etching process (as compared to Ref. [16]) and introduce modifications to the dicing step to reduce contamination. However, they do not demonstrate any correlation between these process changes and improvements in coherence. Such a connection that would have significantly strengthened the manuscript. I can understand the challenges involved in conducting such a study, particularly the time and resources required in an academic setting.

That said, improving the coherence of transmon qubits remains an active area of research across many labs. Moreover, detailed methodology, especially in nanofabrication, is often underreported and undervalued in scientific literature. The authors provide (i) comprehensive information on the processes and equipment used, and (ii) demonstrate a two-fold improvement over previous work. Their transparency and detailing could encourage other research teams to be more forthcoming.

Based on these considerations, I would recommend the manuscript for publication in Nat Comms, despite the lack of correlation between the fabrication changes and coherence improvements.

(Remarks on code availability)

Not applicable.

made.

Point-by-point response to Reviewer 1

Reviewer: I sincerely appreciate the authors' transparency and thorough disclosure of technical details. The community undoubtedly benefits from such open sharing of research. While this spirit of openness should be encouraged, it does not inherently justify publication in a particular journal. Many significant works are described in detail, but that does not automatically qualify them for publication in high-impact journals. The authors have not sufficiently demonstrated substantial novelty in their work. Journals like Nature Communications have well-established standards and criteria that must be adhered to by every submission to maintain fairness and integrity.

For instance, related works such as [9] introduced a new material platform that significantly enhanced the T1 time to 0.3ms in a planar design for the first time, while [10] further improved it to 0.5ms. This current work advances the field with slightly improved T1 time and about a factor of 2 improvement in T2 echo time but still in isolated qubits. This improvement is not substantial, and the underlying reasons are neither explained nor investigated. Even if these improvements were considered significant, they do not necessarily translate to success in multi-qubit scenarios. Not to mention that T2 echo time is not the single metric that matters. I anticipate that the authors will encounter challenges in achieving the same metrics when qubits are interconnected within a realistic and scalable architecture.

Regrettably, I must recommend rejection once more. Although I still believe this work deserves publication, it may be better suited to a more specialized journal. Before resubmission, there are several critical points that the authors need to address:

Response: We thank the reviewer for the review of the manuscript and for noting the importance of open science in our manuscript. As we mentioned before, the coherence times are expected to improve in steps of a factor of two rather than in steps of a factor of ten. In addition, the feedback we have received from the scientific community supports our view of the significance of the results. It is reasonable, in our view, that these results are achieved first with single-qubit systems. Even so, our detailed description of the techniques would help the community scale these results for more practical architectures.

Specific comments:

Reviewer: As highlighted by Reviewer #2, the transmon qubits employed in this study have a relatively low E_J/E_C ratio, which implies reduced sensitivity to dielectric loss. This could be an alternative explanation for the long T1 times, potentially overshadowing the contributions of material and fabrication factors.

Response: We understand the reviewer's concern about the low E_J/E_C ratio. It is true that we do not know exactly how much of the long coherence times can be attributed to material and fabrication factors. However, not only are the T1 times long, but the T2 echo times are as well.

Reviewer: It is well understood that low E_J/E_C ratios result in strong charge dispersion, which likely leads to shorter Ramsey times. The authors should not omit the Ramsey results, as they are crucial for a comprehensive understanding of the work.

Response: We agree that data on the Ramsey experiments would yield additional information, but unfortunately this data is not available for these devices. However, the lack of these data does not change the conclusions of our work.

Reviewer: This study includes measurements of tunable qubits. It is unclear why the authors did not measure qubit lifetime at varying frequencies. Such characterizations are standard in the field and could be completed in a day or two, even without rapid flux biasing, contrary to the authors' claim of it taking a year.

Response: We understand the reviewer's interest in the frequency dependence of the qubit lifetime. However, this aspect was not the focus of our study.

Reviewer: I strongly recommend including data on the thermal excitation of qubits and resonators. These metrics are essential for a thorough understanding of coherence times.

Response: During the measurements, we did not record the thermal excitation of the system, which is why it has not been included in the manuscript.

Reviewer: By addressing these points, the authors can significantly strengthen their manuscript and enhance its potential for successful publication in a suitable journal.

Response: We thank the reviewer for these suggestions which we will take into account in our future work.

Point-by-point response to Reviewer 2:

Reviewer: The authors have carefully addressed the comments. The additional texts and supplementary note illustrate the scope of the manuscript and the implications of the presented results. I have no further comments, except for the following specific point – the expression in line 174 may warrant attention from the authors: "a high vacuum ($<10^{-7}$ mbar) is pumped again to the load-lock chamber".

Response: We thank Reviewer 2 for their feedback. We have revised the specific sentence mentioned.

Point-by-point response to Reviewer 3:

Reviewer: I have broadly reviewed the manuscript submitted by Tuokkola et al. The authors present a detailed methodology for fabricating high-quality transmon qubits.

From a technical standpoint, I find that the peer-review process has improved the quality of the manuscript to an acceptable level. However, the primary concern remains on the novelty of the work, as it is essentially a methodology-focused study.

A previous study (Ref. [16] by Kippenberg et al., Nat. Comm. 525,15,3950(2024)) reported a similar fabrication technique. In the present study, the authors employ a different combination of gases in their etching process (as compared to Ref. [16]) and introduce modifications to the dicing step to reduce contamination. However, they do not demonstrate any correlation between these process changes and improvements in coherence. Such a connection that would have significantly strengthened the manuscript. I can understand the challenges involved in conducting such a study, particularly the time and resources required in an academic setting.

That said, improving the coherence of transmon qubits remains an active area of research across many labs. Moreover, detailed methodology, especially in nanofabrication, is often underreported and undervalued in scientific literature. The authors provide (i) comprehensive information on the processes and equipment used, and (ii) demonstrate a two-fold improvement over previous work. Their transparency and detailing could encourage other research teams to be more forthcoming.

Based on these considerations, I would recommend the manuscript for publication in Nat Comms, despite the lack of correlation between the fabrication changes and coherence improvements.

Response: We thank Reviewer 3 for their comments and for their interest in research on the coherence of superconducting qubits. We also appreciate the reviewer's understanding that conducting a correlation study is challenging for practical reasons. However, we aim to continue exploring this topic in the future including the correlation study to our research. We further thank the reviewer for recognizing the value of a transparent and detailed description of the methods in our manuscript.